# Continuous Microalgal Cultivation for Antioxidants Production

**DOI:** 10.3390/molecules25184171

**Published:** 2020-09-11

**Authors:** Jenny Fabiola López-Hernández, Pedro García-Alamilla, Diana Palma-Ramírez, Carlos Alfonso Álvarez-González, Juan Carlos Paredes-Rojas, Facundo J. Márquez-Rocha

**Affiliations:** 1Academic Division of Biological Sciences, Universidad Juárez Autónoma de Tabasco, Villahermosa, Tabasco 86150, Mexico; jflh23@hotmail.com (J.F.L.-H.); alvarez_alfonso@hotmail.com (C.A.Á.-G.); 2Academic Division of Agriculture and Livestock Science, Universidad Juárez Autónoma de Tabasco, Centro, Tabasco 86280, Mexico; pedro.garciaa@ujat.mx; 3Mexican Center for Cleaner Production, CDMX, Instituto Politécnico Nacional, Ciudad de México 07738, Mexico; dpalmar@ipn.mx (D.P.-R.); paredesrojasjc@gmail.com (J.C.P.-R.); 4Mexican Center for Cleaner Production, Tabasco Unit, Instituto Politécnico Nacional, Cunduacán, Tabasco 86691, Mexico

**Keywords:** antioxidants, phenols, tocopherols, phycocyanin, continuous cultivation, microalgae

## Abstract

Microalgae, including cyanobacteria, represent a valuable source of natural compounds that have remarkable bioactive properties. Each microalga species produces a mixture of antioxidants with different amounts of each compound. Three aspects are important in the production of bioactive compounds: the microalga species, the medium composition including light supplied and the photobioreactor design, and operation characteristics. In this study, the antioxidant content and productivity performance of four microalgae were assessed in batch and continuous cultures. Biomass productivity by the four microalgae was substantially enhanced under continuous cultivation by 5.9 to 6.3 times in comparison with batch cultures. The energetic yield, under the experimental conditions studied, ranged from 0.03 to 0.041 g biomass kJ^−1^. Phenols, terpenoids, and alkaloids were produced by *Spirulina*
*platensis*, *Isochrysis*
*galbana*, and *Tetraselmis*
*suecica*, whereas tocopherols and carotenoids were produced by the four microalgae, except for phycocyanin and allophycocyanin, which were only produced by *S. platensis* and *Porphyridium*
*cruentum*. The findings demonstrate that the continuous cultivation of microalgae in photobioreactors is a convenient method of efficiently producing antioxidants.

## 1. Introduction

Microalgae are a promising group of microorganisms that produce novel compounds, known as secondary metabolites. Value-added microalgal products are associated with the health and cosmetics industries, unrestricted products produced for medical care, and the aquaculture industry [1,2]. Cure from cancer is the goal of cancer therapy. Many deaths in women are due to cervical, stomach, and breast cancers; the demand for anticancer agents from natural sources has increased for antitumor therapy due to the less severe or lack of secondary effects of these treatments [3]. Various microalga species have been reported to produce nutraceuticals, renewable-energy-related compounds, β-carotene, astaxanthin, polyunsaturated fatty acids (PUFAs), and biocolorants; all these compounds are accumulated during growth and are produced for different applications such as health- and food-based products [1], wastewater treatment, aquaculture, etc. [2,4].

The microalgae are primary marine producers consumed by many other marine organisms throughout the food chain. Recent studies showed that some species of algae contain large amounts of antioxidants, such as vitamin E, pigments, and phenolic compounds. Natural antioxidants found in algae may play important roles against various diseases and aging processes by protecting the cells against oxidative damage, including DNA and protein oxidative damage, through high scavenging activity [5,6]. The oxidative mechanism involves an important flow of free radicals following a chain of reactions that occurs in all photosynthetic microorganisms. Oxidation reactions produce free radicals that eventually cause oxidation of key molecules, such as protein and DNA breakage, cell damage, or cell death. For this reason, it is important to maintain a cellular balance of free-radical intermediates with antioxidants produced by the cells, especially those that remove the free-radical intermediates, preventing harmful oxidative reactions [7]. Microalgae produce free-radical scavenging compounds, such as phenols, vitamins, pigments, and enzymes. These compounds can also exhibit antibacterial, antitumor, anticarcinogenic, and anti-inflammatory activities in cells [8]. Red, brown, and green algae, as well as cyanobacteria, display high radical scavenging activities [9]. The main antioxidants found in algae are vitamins C and E [10], carotenoids (β- and α-carotene, zeaxanthin, and neoxanthin), chlorophylls, [11,12], and phenols [13]. The tocopherols show antioxidant activity, providing an important function in the electron transport system, which is a key process in photosynthetic organisms. Other important scavenging elements include enzymes that have antioxidant activity, which are divided into two categories: primary and secondary defense. The primary defense enzymes neutralize free radicals; these include enzymes such as glutathione peroxidase, catalase, and superoxide dismutase. In coordination, they neutralize highly reactive species to less reactive species. The second category comprises enzymes that drive electronic flow during metabolism, e.g., the glutathione reductase reduces glutathione to its reduced form and this recycling continues neutralizing free radicals. Glucose-6-phosphate regenerates NADPH from its oxidized form. Other important nonenzymatic antioxidants include vitamins (A, C, and E), cofactors (Q10), minerals (selenium, zinc), peptides (glutathione), and phenolic and nitrogenate compounds [14]. Metabolites such as phenols, folate, folic acid, and other pigments can produce beneficial effects on health related to hypercholesterolemia, cardiovascular diseases, viral infections, and cancer [7]. Plant antioxidants include a variety of compounds, including tocopherols, ascorbic acid, resveratrol, flavonoids, flavanol, and flavanones, but plants are not the only organisms that produce these compounds; microalgae, as photosynthetic organisms, produce many of these and other compounds with antioxidant properties [15,16]. One advantage of producing antioxidants from microalgae is its cultivation versatility and many technological improvements can be applied since strain selection improving the bioprocesses is important for enhancing a product.

Phycobiliproteins (PBPs) are photosynthetic pigments found in cyanobacteria, rhodophytes, cryophytes, and glaucophytes. PBPs include phycocyanin (PC), allophycocyanin (APC), and phycoerythrin (PE). PC is a fluorescent protein with a prosthetic group (bilins) linked to cysteine residues and a water-soluble compound, with a molecular weight of 586.7 g mol^−1^. Phycocyanin is a natural product with important productive and bioactive advantages for food quality and in the pharmaceutical and cosmetic industries. The use of PC has been recently reported as a natural colorant and an important antioxidant protector. The health and nutraceutical effects of PC include neuroprotective, anti-inflammatory, anticancer, and hepatoprotective properties, which are properties linked to the oxidative mechanism in the photosynthesis system in vivo [17]. Santiago et al. demonstrated that PC has potent antioxidant and synergic activities connected to effectively scavenging hydrogen peroxide [18]. The chemical structure of its chromophore is similar to biliverdin, which protects against oxidative stress and normalizes renal oxidative stress markers and the expression of NAD(P)H oxidase components. PC inhibited NADPH independent of the superoxide production in cultured renal mesangial cells [19]. Other metabolites, such as polyphenols, folate, folic acid, and other pigments, can beneficially affect health, including hypercholesterolemia, cardiovascular diseases, viral infections, and cancer [7]. Chen et al. reported an engineering strategy to simultaneously produce PC with a productivity of 0.13 g L^−1^ d^−1^ and a CO_2_ consumption of 1.58 g L^−1^ d^−1^ with *Spirulina platensis* [20].

Two kinds of photobioreactors exist that produce microalgae biomass: open and closed photobioreactor designs. The dynamics of the cultures allow them to be produced either in batches or continuously, working under natural or artificial light irradiance. However, open systems have two disadvantages: contamination with bacteria and light supply. Additionally, the general cell density is low and only few microalgae species are suitable for this production mode due to contamination. Closed systems are an alternative to producing highly valuable compounds such as antioxidants. The control of all operational parameters is more easily accomplished in closed systems, but costs are high. One alternative to enhancing biomass density and production is operating photobioreactors during continuous production. One difficulty to be overcome is the low growth rate of microalgae that may cause complications in achieving continuous cultivation before washout of the cells [17]. In the present study, antioxidants production by microalgae was assessed in continuous mode operation in a photobioreactor (PBR).

## 2. Results

### 2.1. Growth Analyses in Batch and Continuous Cultivation

The growth profiles were analyzed in the batch and continuous cultures of four microalgae. The growth performance of four microalgae in batch cultivation is shown in Figure 1. Specific growth rates were calculated by the Gompertz growth equation, which were found to be 0.53, 0.49, 0.51, and 1.1 d^−1^ for *Spirulina platensis*, *Isochrysis galbana*, *Tetraselmis suecica*, and *Porphyridium cruentum*, respectively. Biomass concentration and biomass productivity values are summarized in Table 1. The biomass concentrations in batch cultures were 3, 1.5, 1.8, 2.1 g L^−1^ for *S. platensis*, *I. galbana*, *T. suecica*, and *P. cruentum*, respectively (Table 1). The growth of microalgae depends on the culture conditions such as medium composition, hydrodynamics in the bioreactor, and light irradiation that reaches every cell. By comparing all data of the Gompertz growth parameters among all microalgae, μ (growth specific rate) and maximum biomass density were different for all strains (Table 1).

The batch cultures reached maximum cell concentrations over nine days of cultivation; maximum cell concentration indicates the change from batch to continuous cultivation. The dilution rate, which corresponds to the growth rate in continuous culture, was fixed at 0.6 d^−1^ for *S. platensis*, *I. galbana*, *T. suecica*, and *P. cruentum*. Different light intensities were supplied (16.27, 21.26, 30.42, and 39.63 kJ d^−1^). The biomass concentrations of the four microalgae increased as light intensity increased in continuous culture (Table 2). The biomass concentration under all light intensities in continuous cultures were significantly different for all microalgae used. Biomass concentration in descending order at a light intensity of 39.63 kJ d^−1^ was: 3.35 > 2.63 > 1.93 > 1.58 g biomass L^−1^ for *S. platensis*, *P. cruentum*, *T. suecica*, and *I. galbana*, respectively (Table 2), in comparison with biomass in bath cultures: 3.0 > 2.1 > 1.8 > 1.5 g L^−1^, respectively, for the same light irradiance (39.63 kJ d^−1^).

Biomass productivity (XVD (g biomass L^−1^ d^−1^), where X is biomass (g), V is working volume (L^−1^), and D is dilution rate (d^−1^)) obtained in terms of total light supplied (I_0_A (kJ d^−1^) energy supplied from light to a surface area per unit of time) is shown in Figure 2. The productivity of biomass at a light intensity of 39.63 kJ d^−1^ in batch culture was 0.34, 0.24, 0.2, and 0.16 g biomass L^−1^ d^−1^ (Table 1) and 2.01, 1.58, 1.16, and 0.95, g biomass L^−1^ d^−1^, for *S. platensis*, *P. cruentum*, *T. suecica*, and *I. galbana*, respectively, in continuous cultivation (Figure 2). Comparison of the biomass productivity (XVD) though linear regression analysis of each microalga showed statistically significant differences with increasing light intensity. The productivity increase of biomass concentration was 5.9–6.6 times higher in continuous cultures, which may support the final quantity of antioxidants produced, as shown below. The energetic yield of the produced biomass for the supplied light energy varied according to the strain: 0.041, 0.037, 0.033, and 0.030 g biomass produced per kJ for *S. platensis*, *P. cruentum*, *T. suecica*, and *I. galbana*, respectively, and a yield of conversion efficiency (Y_ce_, luminous energy to chemical energy) of 1.04, 0.95, 0.83, and 0.75, respectively (–ΔHo × Y_kcal_; enthalpy change for O_2_ = 106 kcal mol^−1^ O_2_).

### 2.2. Antioxidant Production in Batch Culture

Total phenols, alkaloids, and terpenoids content were measured in *S. platensis*, *I. galbana*, and *T. suecica* during batch cultivation (Table 3). All compounds were produced by the microalgae in significantly different amounts (*p* < 0.05), except for alkaloids between *I. galbana* and *T. suecica*, where no significant differences were observed. Phenol, alkaloids, and terpenoids were more abundant in *S. platensis* than the others; these compounds accumulated during growth.

The productivities of phenols, alkaloids, and terpenoids showed significant differences among the three species (Figure 3). The productivity of phenols in *S. platensis* was four times that of *I. galbana* and 1.75 times that of *T. suecica* (Figure 3). Alkaloids and terpenoids also showed significant differences among the three species of microalgae.

Tocopherols (α, δ, and γ), carotenoids, superoxide dismutase (SOD), and phycobiliproteins were recorded during batch production in the four microalgae (Table 4). The total tocopherol amount is reported here as the sum of α, δ, and γ tocopherols in *I. galbana*, *T. suecica*, as well as the sum of α and γ tocopherols in *S. platensis* and *P. cruentum*. In *S. platensis* and *P. cruentum*, δ-tocopherol was not found. A large amount of tocopherol was produced in *P. cruentum*, with α- and γ-tocopherol at 80.5 and 104.2 μg g^−1^, respectively. Durmaz et al. reported 106.3 μg g^−1^ biomass for both α- and γ-tocopherols in *P. cruentum* [21], which is different from the 185 μg g^−1^ obtained in this study. Figure 4 shows the accumulation of PC and APC in *S. platensis* during growth.

The productivities of antioxidants in microalgae are shown in Table 5. The productivity of tocopherols in *S. platensis* and *P. cruentum* showed no significant differences. Carotenoids, PC, APC, and SOD productivities were statistically different in all microalgae.

### 2.3. Antioxidant Concentration and Productivities in Continuous Culture at Different Light Intensities

The concentrations of total phenols, terpenoids, and alkaloids were analyzed in continuous cultures of *S. platensis*, *I. galbana*, and *T. suecica* at different light intensities. Total phenols, alkaloids, and terpenoids concentrations were significantly different among the three microalgae (*p* < 0.05; Table 6). Phenols, alkaloids, and terpenoids content did not significantly change in any of the microalgae with 16.27 and 21.26 kJ d^−1^ of supplied light intensity (Table 6). By increasing light intensity to 30.42 kJ d^−1^ phenols, alkaloids, and terpenoids content changed significantly in every microalga, except for phenols content in *I. galbana*. Phenols, alkaloids, and terpenoids content showed significant differences at 39.63 kJ d^−1^ in comparison with the contents of these compounds at 16.27 and 21.26 kJ d^−1^ in every microalga. A significant difference in phenols content was found in *S. platensis* and *I. galbana* by changing light intensity from 30.42 to 39.63 kJ d^−1^ (Table 6).

The productivity of phenols, alkaloids, and terpenoids in *S. platensis*, *I. galbana*, and *T. suecica* under four different light intensities cultured under continuous cultivation is shown in Figure 5A–C. The three compounds’ productivity increased significantly with increasing light intensity in every microalga. Phenol productivity in *S. platensis* was five times higher than in *I. galbana*, and more than twice that of *T. suecica* at a light intensity of 39.63 kJ d^−1^. Phenol productivity increased more than twice in *S. platensis*, more than four times in *I. galbana*, and three times than *T. suecica* with the change in light intensity from 16.27 to 39.63 kJ d^−1^ (Figure 5A). Alkaloid productivity of *S. platensis* was more than three times that of *I. galbana*, and more than four times that of *T. suecica* (Figure 5B). The alkaloid productivity was 2.1 times higher in *S. platensis*, 4.1 times in *I. galbana*, and more than three times higher in *T. suecica* with increasing the light intensity from 16.27 to 37.63 kJ d^−1^ (Figure 5B). Terpenoids productivity increased significantly with increasing light intensity. The productivity of terpenoids in *S. platensis* was significantly different from the productivity in *I. galbana* and *T. suecica*, whereas productivity of terpenoids in *I. galbana* and *T. suecica* showed no significant differences (Figure 5C).

Tocopherols content was significantly different in the four microalgae analyzed. *T. suecica* and *P. cruentum* showed higher amounts of tocopherols than *S. platensis* and *I. galbana*. In addition, tocopherols concentration increased with light intensity in the four microalgae (Table 7). Tocopherol concentration increased 1.22 times in *P. cruentum* to 1.56 times in *I. galbana* by changing light intensity from 16.27 to 39.63 kJ d^−1^.

Tocopherols productivity in *S. platensis* and *P. cruentum* was not significantly different at 16.27 and 21.26 kJ d^−1^ of light intensity supplied, but significant different when light intensity increased from 21.26 to 39.63 kJ d^−1^. *P. cruentum* showed a productivity increase of more than 1.2 times the productivity of *S. platensis*, 2.66 times the productivity of *I. galbana*, and 1.57 times the productivity of *T. suecica* (Figure 6).

The carotenoids content in the microalgae changed significantly from light intensities of 16.27 to 39.63 kJ d^−1^. Carotenoid content increased 1.5, 1.35, 1.56, and 1.57 times in *S. platensis*, *I. galbana*, *T. suecica*, and *P. cruentum*, respectively. At a light intensity of 39.63 kJ d^−1^, carotenoids content showed no significant differences among the microalgae studied (Table 8).

Carotenoids productivity changed with the biomass productivity for each microalga; *S. platensis* had a productivity at least twice that of *I. galbana* and *T. suecica* and at least 1.3 times that of *P. cruentum* (Figure 7). Carotenoids productivities in *T. suecica* and *I. galbana* were not significantly different at light intensities of 16.27 and 21.26 kJ d^−1^.

The amounts of UA SOD were significantly different among *S. platensis*, *I. galbana*, and *P. cruentum* at different light intensities, but no difference was observed between *S. platensis* and *T. suecica*. SOD units increased significantly by increasing light intensity supplied in each microalga (Table 9).

The SOD UA productivity increased in accordance with biomass productivity with increasing supplied light energy (Figure 8). Soluble protein was found to be 65.7, 43.5, 51.3, and 77.9 mg protein g^−1^ biomass for *S. platensis*, *I. galbana*, *T. suecica*, and *P. cruentum*, respectively. SOD UA productivity was significantly different in all microalgae.

Phycocyanin (PC) and allophycocyanin (APC) were found only in *S. platensis* and *P. cruentum*. Compared with the other antioxidants analyzed in this work, PC and APC content decreased with increasing light intensity (Table 10). PC concentration in *S. platensis* was 5.6 times the concentration in *P. cruentum*, and APC concentration in *S. platensis* was 7.1 times the concentration of APC in *P. cruentum* at a light intensity of 39.63 kJ d^−1^.

The PC production by *S. platensis* was 7.0 times that by *P. cruentum*; APC production of *S. platensis* was 8.9 times that of *P. cruemtum* (Figure 9). Although PC and APC content decreased with increasing light intensity, the production of both pigments increased with increasing light intensity due to biomass productivity.

The extract of *S. platensis* was dissolved in solvent, then the extract was analyzed by FTIR (Table 11). Several signals appeared related to some characteristic vibrational groups. An important vibrational signal, the methylene group (2829–2848 cm^−1^ [22]), appeared at 2835 cm^−1^, which is possibly found in carotenoids and phytosterols molecules such as lycopene, which is a precursor of carotene [23]. Another vibrational signal found at 1230 cm^−1^ was assigned to the phenyl group and 1517 cm^−1^ to the stretching of the flavone phenyl ring; these signals correlate well with antioxidants such as flavone. The vibration band at 1740 cm^−1^ corresponds to the carbonyl group of phytosterols and the signal at 1612–1652 cm^−1^ to the C=O group, which are signals assigned to ketones found in flavonoids.

## 3. Discussion

The microalgae studied in this work exhibit reliance on antioxidant production on the growth conditions; however, strain species is an important factor. Antioxidant accumulation occurs along with cell growth, playing important roles in microalgal cells.

Phycocyanin has different biotechnological applications in cosmetics, food, and feed industries, analytical biochemistry, and therapeutic applications [19]. Zheng et al. [19] reported PC-protection against nephropathy by inhibiting oxidative stress, perhaps due to the similarity of the PC chemical structure with that of biliverdin. Chen et al. [20] reported a strategy to simultaneously improve biomass and PC productivities in a flat PBR with batch cultivation of 0.74 and 0.125 g L^−l^ d^−1^ under a light intensity of 700 μmol m^−2^ s^−1^ (32.95 × 10^−3^ kJ cm^−2^ h^−1^) [20], which is close to double the 15.1 × 10^−3^ kJ cm^−2^ h^−1^ (320 μmol m^−2^ s^−1^) light intensity used in the present work. Here, the biomass and PC productivities were 2 and 0.4 g L^−1^ d^−1^, respectively, which are 2.7 times the biomass and 3.2 times the PC productivity at 320 mmol m^−2^ s^−1^ compared to the values reported by Chen et al. [20] at 700 μmol m^−2^ s^−1^. Although PC content in *S. platensis* was 235 PC mg g^−1^ biomass at 132 μmol photons m^−2^ s^−1^ and 198 PC mg g^−1^ biomass produced at 320 μmol m^−2^ s^−1^, PC productivity at 320 μmol photons m^−2^ s^−1^ was 1.7 times the PC produced at 132 μmol m^−2^ s^−1^. Zheng et al. [19], in a PBR with a light intensity of 200 μmol m^−2^ s^−1^ in batch culture, obtained 0.39 g L^−1^ d^−1^ biomass productivity for *S. platensis* [19]. Schipper et al. worked with *Leptolyngbya* sp. to produce PC in continuous culture at 80 μmol photons m^−2^ s^−1^. Under these conditions, 86 mg of PC g^−1^ biomass was produced with a productivity of 78.8 mg L^−1^ d^−1^ [24].

Microalgae produce important antioxidants: alkaloids, terpenoids, phenols, tocopherols, carotenoids, phycobilins, and other pigments and enzymes, which are involved in active antioxidant treatments due to their anti-inflammatory, antibacterial, and anticancer, lipid peroxidation activities. *Porphyridium cruentum* produced another promising compound, sulfate-polysaccharide, which has either anti-inflammatory or antiallergic properties. α-tocopherol is the most abundant with the highest antioxidant activity in terrestrial plants; it functions in the electron transport reactions and cell membrane stabilization functions [25]. Animals are unable to synthesize tocopherols, so plant sources are a primary choice for human consumption. Nevertheless, microalgae have become a natural and alternative source of tocopherols, containing other tocopherols that provide remarkable synergistic antioxidant activity [18]. Durmaz et al. [21] reported 55.2 and 51.3 μg g^−1^ biomass of α- and γ-tocopherol, respectively, in *P. cruentum* [21]. The amounts of α- and γ-tocopherols (80 and 104 μg g^−1^ biomass, respectively) in *P. cruentum* found in this study are in agreement with those reported by Durmaz et al. [21]. In addition, δ-tocopherol was found in *T. suecica*, which produces γ-tocopherol in considerable amounts, but also produced α-tocopherols. *Spirulina maxima* produces α- and γ-tocopherols [18], but the amounts are lower than in *S. platensis* in this study.

Carotenoids and their derivative compounds bond to lipoproteins and play an important role as antioxidants during photosynthesis; they have been associated with potential protection against lipid peroxidation, DNA and protein oxidation, and in cancer prevention [8]. Photosynthetic organisms contain carotenoids; particularly in microalgae, α-, and β-carotenes are present, whereas other microalgae are important axthasantin producers. The carotenoid content in microalgae rarely surpasses 10% and they are present as a mixture of carotenoids. In *P. cruentum*, Rebollo et al. [26] found 1.0 mg g^−1^ biomass, whereas Santiago et al. reported 4.6 mg g^−1^ biomass for β-carotene in *T. suecica* [18].

SOD has been found in *S. maxima*, *Nannochloropsis* sp., *T. suecica*, *Chaetoceros* sp., *Synechoccocus*, and *P. cruentum*, with different amounts in each microalga. In *T. suecica* and *P. cruentum*, activities of 20.4 and 58 UA mg^−1^ protein were measured, respectively [18], in comparison with 54.6 and 74.5 UA mg^−1^ protein reported in this work, respectively. The culture conditions may affect the formation of antioxidant compounds, in particular SOD; this enzyme may be induced in response to the increase or demand to drive reactive oxygen species generated by an increase in photosynthesis activity.

The FTIR spectra of *S. platensis* extracts showed vibrational signals related to some antioxidant molecules, such the signal at 2835 cm^−1^ assigned to the methylene group, at 1230 cm^−1^ to the phenyl group, the stretching at 1517 cm^−1^ assigned to the flavone phenyl ring, and the vibrational signals at 1612–1652 cm^−1^ assigned to the C=O ketone groups found in flavonoids.

Microalgae extracts containing a mixture of antioxidants have the capacity to scavenge superoxide anions, hydrogen peroxides, and hydroxyl radicals [18]. The antioxidant machinery involves enzymatic and nonenzymatic antioxidants. These extracts also prevent oxidative damage by efficiently neutralizing free radicals, which includes catalase by the conversion of hydrogen peroxide into water and molecular oxygen. Other enzymes do not neutralize free radicals directly but create a reducing environment. Nonenzymatic antioxidants include vitamin E, cofactor Q_10_, peptides (glutathione), phenols, and pigments [7]. All these antioxidants are synthesized in microalgae in different combinations and concentrations, highly dependent on strain, culture medium composition, time of harvesting, bioreactor geometry, and irradiated surface. The contents of antioxidants found in microalgae provide the possibility to focus efforts on obtaining reliable sources of antioxidants for production in biofactories.

## 4. Materials and Methods

### 4.1. Microalgae Strains, Culture Medium and Growth Conditions

The microalgae *Porphyridium cruentum*, *Isochrysis galbana*, *Tetraselmis suecica*, and *Spirulina platensis* were acquired from our own culture collection of microalgae. For *I. galbana*, *T. suecica*, and *P. cruentum*, medium “f” [27] of Guillard and Ryther was used. *S. platensis* was grown on SOT standard medium [28]. Cultivation was performed in a jar-fermenter (Applikon Biotechnology BV, JG Delft, The Netherlands) equipped with 2 Rushton impellers 60 mm in diameter and a working volume of 1.0 L. The top surface of the reactor (illuminated surface) had a surface area of 109.36 cm^2^ and diameter of 11.8 cm. Light (0.149, 0.194, 0.278, and 0.36 Jcm^−2^ d^−1^, with an irradiated surface of 109.36 cm^2^, producing I_0_A = 16.27, 21.26, 30.42, and 39.63 kJ d^−1^, respectively) was supplied on the top of the bioreactor. The reactor was operated at a 30 ± 2 °C, agitation 200 rpm, and with an initial pH of 7.5 for *Porphyridium cruentum*, *Isochrysis galbana*, and *Tetraselmis suecica*, and pH 8.5 for *Spirulina platensis*. The illumination was supplied with white light LED lamps on the top of the reactor. The photobioreactor was covered with aluminum foil, assuming that light filled the entire volume of the culture. The bioreactor was connected to a reservoir containing fresh medium; the flow rate was controlled by a valve. The fresh medium was fed into the bioreactor and the broth exit valve was open at the same flow rate. Dilution rate (D) was achieved after at least three days without changes in the cell density in the bioreactor, then the culture was considered in continuous mode or in a steady state. Cells were harvested by centrifugation at 12,000× *g* for 10 min at 4 °C, then the collected biomass was kept at –20° C if not used immediately. (Note: Conversions used: 1 lux = 1.55 μmol m^−2^ s^−1^; 1 cd = 10.76 lux; 1 cd = 7.85 × 10^−6^ kJ cm^−2^h^−1^), (I_0_A is defined as the energy from light supplied to a surface area per unit of time). 

### 4.2. Quantitation of Phenols, Alkaloids, and Terpenoids

*S. platensis*, *I. galbana*, and *T. suecica* biomass (0.1–0.5 g) were suspended in 5.5 mL of 0.01 M phosphate buffer, pH 6.5, and 0.15M NaCl. The suspension was frozen at −20 °C, thawed and sonicated, and then left at 5 °C overnight. The suspension was centrifuged at 12,000× *g* for 15 min at 4 °C; the supernatant was considered crude extract. Total alkaloids was measured as follows: The crude extract was mixed with chloroform (1:1 *v*/*v*) and the chloroform phase was separated and concentrated. The concentrate was resuspended with 1 mL of chloroform, and 6.26 mL of H_2_SO_4_ (0.02N) was added; chloroform was eliminated. Some drops of methyl red indicator were added into the solution, then the mixture was titrated with NaOH (0.02 N). Each mL of NaOH (0.02 N) added was equivalent to 5.78 mg of alkaloids. Total phenols was measured as follows: 0.1 mL of the extract was mixed with 2.8 mL of deionized water, 2 mL of sodium carbonate (2%), and 0.1 mL of Folin–Ciocalteu reagent (50%). The mixture was incubated for 30 min at room temperature and the absorbance of the mixture was recorded at 750 nm in a spectrophotometer [29]. Ferulic acid was used as the standard. For the determination of total terpenoids, 5 mL of extract was placed in an oven at 100 °C for 1 h. After cooling, 5 mL of freshly prepared vanillin reagent (0.7% in 65% H_2_SO_4_) was added. The tubes were heated at 60 °C in a water bath for 1 h. After cooling on ice, the absorbance was measured at 473 nm. Saponin was used as the standard. For FTIR analysis, an FTIR spectrometer (Frontier model, PerkinElmer, Waltham, MA, USA) was used; the sample was redissolved with a mixture of acetonitrile/methanol (70:30).

### 4.3. Tocopherols and Carotenoids Determination

Biomass (0.1–0.15 g) was mixed with a 2 mL solution containing 60% ethanol, pyrogallol (6%), and 0.87 mL KOH (60%) under N_2_ atmosphere, then it was sonicated. The mixture was incubated for 1 h at 70 °C in the dark, then the mixture was chilled on ice and 3.13 mL (NaCl 2%) was added and then extracted with n-hexane (1:1 *v*/*v*). The n-hexane was eliminated under N_2_. The concentrated extract was suspended in 0.7–1.0 mL of methanol and filtered through a 0.45 μm membrane, the extract was filtered and stored at −70 °C. The methanolic extract was used to measure tocopherols in HPLC [18]. Twenty microliters of the extract were passed through a Zorbax XDB-C8 (4.6 mm × 150 mm, 5 μm; Agilent Technologies, Santa Clara, CA, USA) column and eluted with an isocratic solution of methanol/water (95:05) at a flow rate of 1.0 mL min^−1^. A fluorescence detector using an excitation λ_292_, an emission wavelength of λ_340_, and a UV detector at λ_340_ were used to measure the tocopherols content [18]. Commercial α, δ, and γ-tocopherols served as the standards for quantification and detection. Total carotenoids were measured at 480 nm (A_1%_ = 2500 absorption coefficient) of absorbance in a UV-Vis spectrophotometer [30].

### 4.4. Superoxide Dismutase Activity Determination

We mixed 0.05 to 0.1 g of biomass with 1 mL phosphate buffer (0.2 M, pH 8.0, 5 mM EDTA, 1 mM DTT), sonicated for 2 min, then centrifuged at 12,000× *g* at 4 °C for 30 min. The supernatant was used for measurements or stored frozen. For SOD activity determination, the Bioxytech SOD-525 assay kit (Oxis International, Foster, CA, USA) was used. The reaction was followed at 525 nm at 30 °C for 1 min. The reaction contained 50 μg protein mL^−1^ of extract. Commercial SOD was used as the standard. Protein content was measured using Bio-Rad reagent (Bio-Rad Laboratories, Inc., Berkeley, CA, USA) at 595 nm in a spectrophotometer [18].

### 4.5. PC and APC Determinations

*S. platensis* and *P. cruentum* biomass (0.1–0.5 g) was suspended in 5.5 mL of potassium phosphate buffer 0.01 M, pH 6.5, and 0.15M NaCl. The suspension was frozen at −20 °C, thawed, and sonicated, then left at 5 °C overnight. The suspension was centrifuged at 12,000× *g* for 15 min at 4 °C. The supernatant contained phycocyanin–allophycocyanin extract. Phycocyanin and allophycocyanin concentrations were calculated according to Bennet and Bogard [31] by measuring the absorbance at 615 and 652 nm in a spectrophotometer and applying the following equations (data in mg mL^−1^): where OD = optical density
$$\mathrm{PC}\;=\;\frac{OD_{615}-0.474\left(OD_{652}\right)}{5.34}$$
$$\mathrm{APC}\;=\;\frac{OD_{652}-0.208\left(OD_{615}\right)}{5.09}$$

### 4.6. Statistical Analyses

GraphPad Prisma software version 8.4.3 (GraphPad Software, San Diego, CA, USA) was used for all analyses. For growth, the Gompertz model was used to calculate growth parameters. For biomass productivity at different light intensities, a linear regression analysis was applied and we compared the 4 strains. To compare the concentration and productivity of all strains at different light intensities, Tukey’s multiple comparison test was applied (*p* < 0.05).

## 5. Conclusions

Microalgae, including cyanobacteria, are manipulatable organisms that grow in photobioreactors, where the medium composition, light intensity, reactor design, reactor operation, metabolism, and microalga species are important factors affecting the production of compounds synthesized by the microalgae. In this work, well-known microalgae species that produce secondary metabolites were assayed for the production of different compounds that have remarkable antioxidant activity. We found that the microalgae produce a mixture of antioxidants in different amounts depending on the species, light supplied, and bioreactor operation. Phenols, alkaloids, terpenoids, tocopherols, carotenoid, PC, and APC content, including an enzyme, SOD, were produced by the microalgae tested. Continuous bioreactor operation substantially improved biomass and antioxidant productivity in comparison with batch cultivation in the same bioreactor under the same conditions, except that continuous cultivation produces the same amount of biomass and secondary metabolites every day over nine days of cultivation as achieved in batch cultures. Continuous cultivation drives the microalgal cell metabolism of biomass and metabolites synthesis in the bioreactor; the mathematical description of continuous cultivation is referred to as a steady state, where biochemical pathways constantly dynamically synthesize molecules. Antioxidants are produced naturally as a mixture. We deduced that the combination of continuous culture with the correct nutrient limitation will improve or enhance the production of a specific antioxidant, but this may limit the production of other antioxidants.

## Figures and Tables

**Figure 1 molecules-25-04171-f001:**
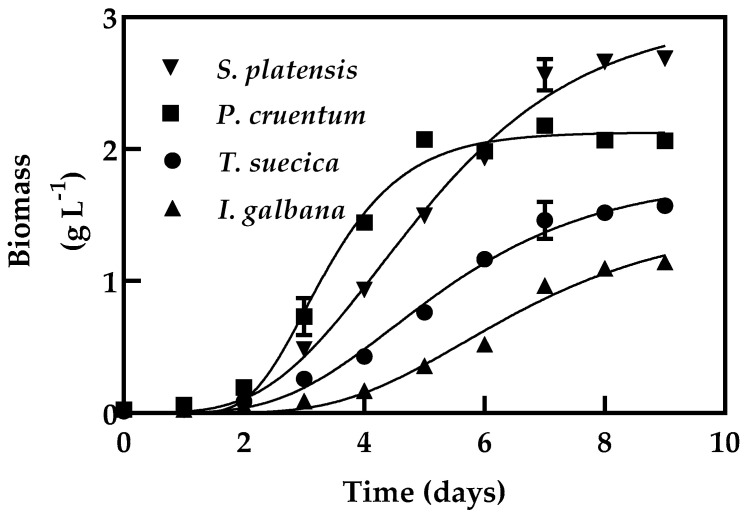
Growth of microalgae in batch cultures. Microalgae were cultivated in batch operation mode in a photobioreactor at 200 rpm, at 30 °C, 39.63 kJ d^−1^ of light supplied, and 1 L working volume. Each point is the mean of 3 replicates.

**Figure 2 molecules-25-04171-f002:**
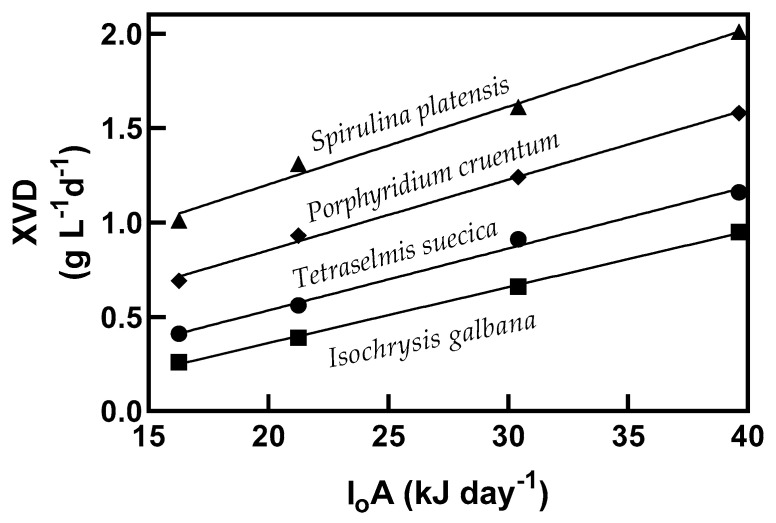
The biomass produced in steady state condition at D (dilution rate) = 0.6 d^−1^ for each energy supplied in continuous cultures of *S. platensis*, *P. cruentum*, *T. suecica*, and *I. galbana.*

**Figure 3 molecules-25-04171-f003:**
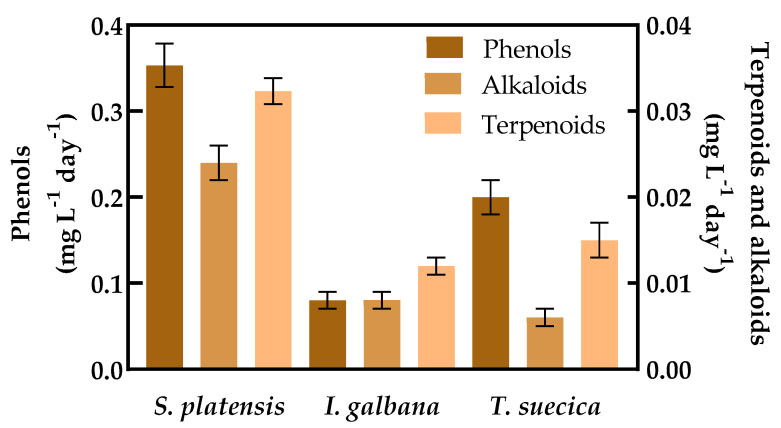
Antioxidant productivities in batch cultures. Phenols, alkaloids, and terpenoids productivities in *S. platensis*, *I. galbana*, and *T. suecica* in batch cultures under a supplied light intensity of 39.63 kJ d^−1^ for 9 days of cultivation. Values are the means of 3 replicates (*p* < 0.05).

**Figure 4 molecules-25-04171-f004:**
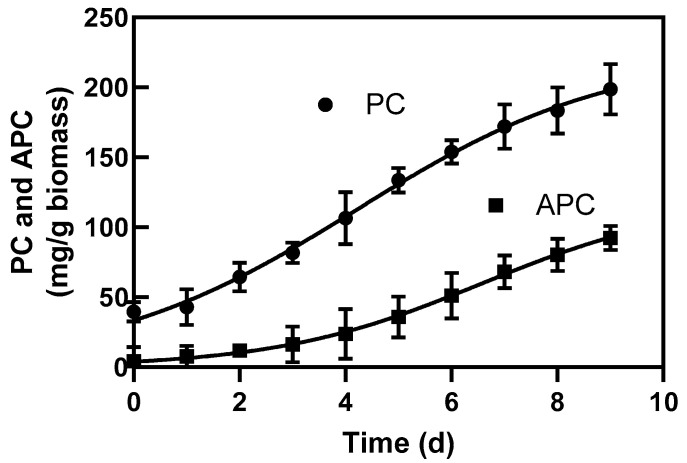
PC and APC formation during growth. Phycocyanin (PC) and allophycocyanin (APC) formation along with the growth of *S. platensis* in batch cultures supplied a light irradiance of 39.63 kJ d^−1^ during 9 days culture.

**Figure 5 molecules-25-04171-f005:**
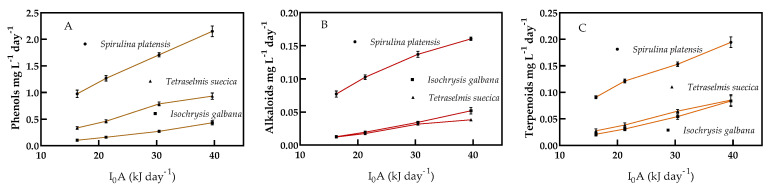
Phenols, alkaloids, and terpenoids productivities under continuous culture. (**A**) Phenols; (**B**) Alkaloids; (**C**) Terpenoids. At supplied light intensities (X axis) of 16.27, 21.26, 30.42, and 39.63 kJ d^−1^. Each point is the means of 3 replicates (*p* < 0.05).

**Figure 6 molecules-25-04171-f006:**
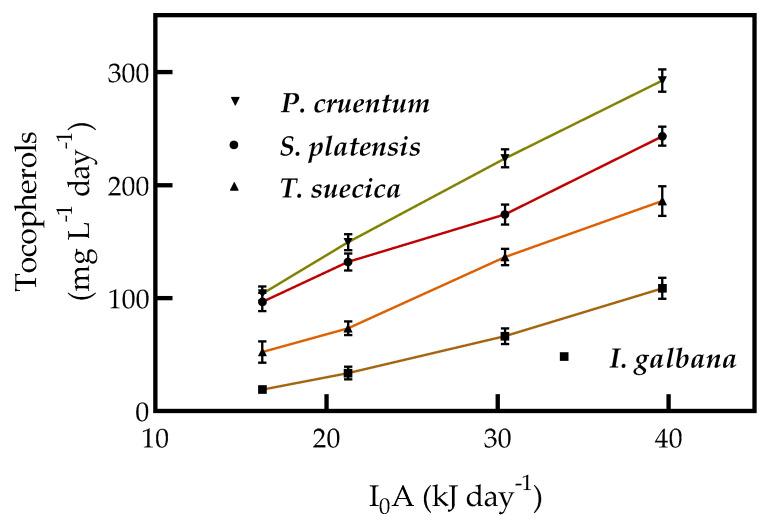
Tocopherols productivity in continuous cultures. Tocopherols productivity by *S. platensis*, *I. galbana*, *T. suecica*, and *P. cruentum* in continuous culture with different supplied light intensities (16.27, 21.26, 30.42, and 39.63 kJ d^−1^). Each point represents the mean of 3 replicates (*p* < 0.05).

**Figure 7 molecules-25-04171-f007:**
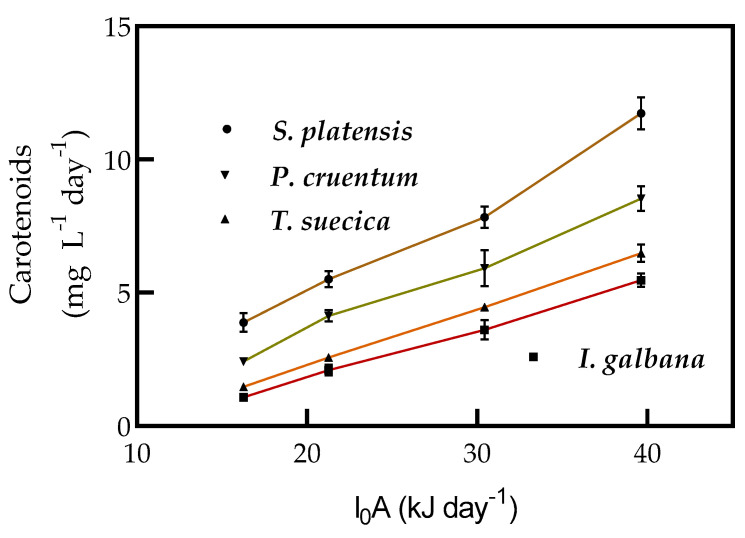
Carotenoids productivity of *S. platensis*, *I. galbana*, *T. suecica*, and *P. cruentum* in continuous culture under supplied light intensities of 16.27, 21.26, 30.42, and 39.63 kJ d^−1^. Each point indicates the mean of 3 replicates (*p* < 0.05).

**Figure 8 molecules-25-04171-f008:**
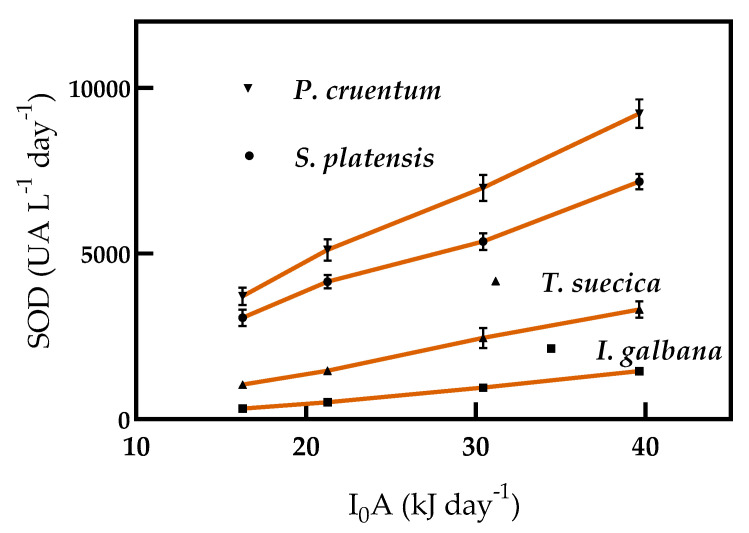
SOD productivity by *S. platensis*, *I. galbana*, *T. suecica*, and *P. cruentum* in continuous culture with different supplied light intensities. Each point indicates the mean of 3 replicates (*p* < 0.05).

**Figure 9 molecules-25-04171-f009:**
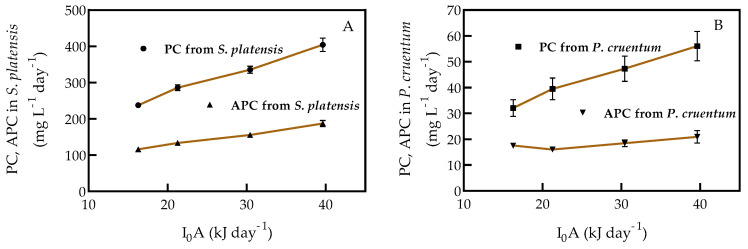
PC and APC production in (**A**) *S. platensis* and (**B**) *P. cruentum* under continuous cultures at different light intensities. Each point indicates the mean of 3 replicates (*p* < 0.05).

**Table 1 molecules-25-04171-t001:** Growth parameters. Values obtained in bath cultivation for 9 days’ growth.

Strain	μ	Biomass	Biomass Productivity
	**d^−1^**	**g L^−1^**	**g biomass L^−1^ d^−1^**
*Spirulina platensis*	0.53 ^a^	3.02 ^e^	0.34 ^i^
*Isochrysis galbana*	0.49 ^b^	1.45 ^f^	0.16 ^j^
*Tetraselmis suecica*	0.51 ^c^	1.78 ^g^	0.20 ^k^
*Porphyridium cruentum*	1.10 ^d^	2.13 ^h^	0.24 ^l^

Note: Different letters indicate significant differences; 39.63 kJ d^−1^ of light supplied. Values are the means of 3 replicates (*p* < 0.05).

**Table 2 molecules-25-04171-t002:** Biomass accumulation: biomass concentration of microalgae at different light intensities supplied to continuous cultures.

I_0_A	*S. platensis*	*I. galbana*	*T. suecica*	*P. cruentum*
(kJ d^−1^)	g biomass L^−1^			
16.27	1.68 ^a^	0.43 ^e^	0.68 ^i^	1.15 ^m^
21.26	2.18 ^b^	0.65 ^f^	0.93 ^j^	1.55 ^n^
30.42	2.68 ^c^	1.10 ^g^	1.52 ^k^	2.07 ^o^
39.63	3.35 ^d^	1.58 ^h^	1.93 ^l^	2.63 ^p^

Note: Different letters indicate significant differences in the means of 3 replicates (*p* < 0.05). I_0_A is defined as the energy supplied from light to a surface area per unit of time.

**Table 3 molecules-25-04171-t003:** Antioxidants concentration in batch cultures. Phenols, alkaloids, and terpenoids content in *S. platensis*, *I. galbana*, and *T. suecica* in batch cultures with a supplied light intensity of 39.63 kJ d^−1^ in 9 days’ culture.

Strain	Phenols	Alkaloids	Terpenoids
	**mg g^−1^ biomass**		
*S. platensis*	1.02 ^a^	0.07 ^d^	0.094 ^f^
*I. galbana*	0.47 ^b^	0.05 ^e^	0.081 ^g^
*T. suecica*	0.83 ^c^	0.03 ^e^	0.075 ^h^

Note: The same letter in each column indicates no significant differences in the means of 3 replicates (*p* < 0.05).

**Table 4 molecules-25-04171-t004:** Antioxidant concentrations in batch culture: tocopherols, carotenoids, phycocyanin (PC), allophycocyanin (APC), and superoxide dismutase (SOD) content in *S. platensis*, *I. galbana*, *T. suecica*, and *P. cruentum* in batch cultures supplied with a light energy of 39.63 kJ d^−1^ for 9 days of culture.

Strain	Tocopherols	Carotenoids	PC	APC	SOD	SOD
	μg g^−1^		mg g^−1^		UA mg^−1^ protein	UA g^−1^ biomass
*S. platensis*	120.5 ^a^	3.70 ^d^	198 ^g^	92 ^i^	53.9 ^k^	3541.23 ^m^
*I. galbana*	115.5 ^a^	5.13 ^e^	nd	nd	35.3 ^l^	1535.55 ^n^
*T. suecia*	159.8 ^b^	4.71 ^f^	nd	nd	54.6 ^k^	2801.01 ^o^
*P. cruentum*	184.7 ^c^	4.61 ^f^	35.4 ^h^	13.2 ^j^	74.5 ^l^	5803.61 ^p^

The same letter in each column indicates nonsignificant statistical difference in the means of 3 replicates (*p* < 0.05). nd: not determined. UA (activity units).

**Table 5 molecules-25-04171-t005:** Antioxidant productivity in batch cultures. Tocopherols, carotenoids, PC, APC, and SOD productivities in batch cultures with a light irradiance of 39.63 kJ d^−1^ with 9 days of culture.

Strain	Tocopherols	Carotenoids	PC	APC	SOD
	μg L^−1^ d^−1^	mg L^−1^ d^−1^			UA L^−1^ d^−1^
*S. platensis*	40.97 ^a^	1.26 ^d^	67.32 ^h^	31.28 ^j^	1204.02 ^l^
*I. galbana*	18.48 ^b^	0.82 ^e^	nd	nd	245.69 ^m^
*T. suecia*	31.96 ^c^	0.94 ^f^	nd	nd	560.20 ^n^
*P. cruentum*	44.33 ^a^	1.11 ^g^	8.50 ^i^	3.17 ^k^	1392.87 ^o^

The same letter in each column indicates no significant difference in the means of 3 replicates (*p* < 0.05). nd: not determined.

**Table 6 molecules-25-04171-t006:** Phenol, alkaloids, and terpenoids content of three microalgae in continuous cultures at different light intensities.

I_o_ A	*S. platensis*			*I. galbana*			*T. suecica*		
kJ d^−1^	mg compound g^−1^ biomass								
	A	B	C	A	B	C	A	B	C
16.27	0.96 ^a^	0.076 ^d^	0.091 ^f^	0.38 ^i^	0.049 ^k^	0.077 ^m^	0.81 ^o^	0.030 ^q^	0.068 ^s^
21.26	0.98 ^a^	0.079 ^d^	0.092 ^f^	0.40 ^i^	0.051 ^k^	0.080 ^m^	0.83 ^o^	0.031 ^q^	0.070 ^s^
30.42	1.06 ^b^	0.085 ^e^	0.095 ^g^	0.41 ^i^	0.053 ^l^	0.083 ^n^	0.87 ^p^	0.035 ^r^	0.073 ^t^
39.63	1.12 ^c^	0.082 ^e^	0.096 ^g^	0.45 ^j^	0.055 ^l^	0.085 ^n^	0.85 ^p^	0.033 ^r^	0.075 ^t^

A, phenols; B, alkaloids; C, terpenoids. Identical letters in each column indicate no significant statistical difference in the means of 3 replicates (*p* < 0.05).

**Table 7 molecules-25-04171-t007:** Tocopherols concentration. Tocopherols content of the four microalgae in continuous culture at different light intensities.

I_o_ A	*S. platensis*	*I. galbana*	*T. suecica*	*P. cruentum*
(kJ d^−1^)	μg tocopherols g^−1^ biomass			
16.27	95.7 ^a^	74.1 ^e^	129.4 ^i^	151.3 ^m^
21.26	101.1 ^b^	86.9 ^f^	131.2 ^j^	160.6 ^n^
30.42	108.1 ^c^	101.3 ^g^	150.2 ^k^	180.1 ^o^
39.63	120.5 ^d^	115.5 ^h^	159.8 ^l^	184.7 ^p^

All values are statistically different in the means of 3 replicates (*p* < 0.05).

**Table 8 molecules-25-04171-t008:** Carotenoids content of the four microalgae analyzed in continuous culture at different supplied light intensities.

I_o_ A	*S. platensis*	*I. galbana*	*T. suecica*	*P. cruentum*
(kJ d^−1^)	(mg carotenoids g^−1^ biomass)			
16.27	3.9 ^a^	4.3 ^b^	3.6 ^a^	3.5 ^a^
21.26	4.2 ^b^	5.3 ^e^	4.6 ^f^	4.5 ^f^
30.42	4.9 ^c^	5.6 ^d^	4.9 ^c^	5.0 ^c^
39.63	5.9 ^d^	5.8 ^d^	5.6 ^d^	5.5 ^d^

Identical letters in each column indicate no significant statistical difference in the means of 3 replicates (*p* < 0.05).

**Table 9 molecules-25-04171-t009:** UA of SOD in microalgae in continuous cultures at different light intensities.

I_o_ A	*S. platensis*	*I. galbana*	*T. suecica*	*P. cruentum*
(kJ d^−1^)	UA SOD mg^−1^ protein			
16.27	45.7 ^a^	29.1 ^e^	48.8 ^b^	68.2 ^i^
21.26	48.3 ^b^	30.4 ^f^	51.5 ^c^	70.1 ^j^
30.42	50.4 ^c^	33.2 ^g^	53.2 ^d^	71.9 ^k^
39.63	53.9 ^d^	35.3 ^h^	54.6 ^d^	74.5 ^l^

Identical letters indicate no statistic differences in the means of 3 replicates (*p* < 0.05).

**Table 10 molecules-25-04171-t010:** PC and APC accumulation in *S. platensis* and *P. cruentum* in continuous cultures at different supplied light intensities.

I_o_ A	*S. platensis*	*P. cruentum*	*S. platensis*	*P. cruentum*
(kJ d^−1^)	mg PC g^−1^ biomass		mg APC g^−1^ biomass	
16.27	235	47	115	25
21.26	217	42	102	17
30.42	210	38	97	15
39.63	198	35.4	92	13

Different letters indicate statistically significant differences. All values are the means of 3 replicates (*p* < 0.05).

**Table 11 molecules-25-04171-t011:** FTIR analyses of *S. platensis* extracts. Signal vibrations found in the *S. platensis* extract dissolved in solvents.

Wave Number (cm^−1^)	Vibrational Signal	Attributed Chemical	Peak Intensity (Height–Width)
2947	ν(C–H)	methylene	short fine
2835	ν(C–H)	methylene group carotenoids	short fine
1740	Carbonyl group	carbonyl group phytosterols	medium fine
1653	ν(C=O)	ketone	medium fine
1620	ν(C=O)	ketone	medium fine
1517	ν_S_(C=C)	flavone phenyl ring	medium fine
1452	ν(C=C)	aromatic	short medium
1409	ν(C=C)	aromatic	short fine
1230	ν(C–O–C)	phenyl ether linkage	medium fine
1025	ν(C–H), ν(C–O–C)	ether	medium fine
948	ν(C–H)	methylene	medium fine

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
