# Peer review of "Continuous Microalgal Cultivation for Antioxidants Production"

_molecules, 2020, doi:10.3390/molecules25184171_

Round 1
Reviewer 1 Report
The manuscript was based on comparative study of batch and continuous microalgal cultivation modes and their effects on the production of antioxidants.
The manuscript is interesting but requires several adjustments to be suitable for publication:
- Line 40: Authors must explain abbreviations the first time they are used.
- Line 86: in vivo must be in italic form.
- Throughout the manuscript: Authors should check missing italics in the scientific name of microalgae species.
- Line 113 to Line 117: Given informations are suitable to be included in Materials and Methods section; since it describe the experimental design of the conducted batch cultivation mode.
- Line 135 to Line 142: Same remark. Informations about changing cultivation mode of microalgae from batch to continuous mode must be included on Materials and methods section.
- Table 3 and 4: Authors must also indicate significant differences of means within the same column with different superscript letters.
- Table 5: Letters indicating differences must be on superscript form.
- Authors should give the significance of “nd” cited in tables.
- The form and presentation of figures should be improved and revised.
- Authors inserted many tables and figures in the manuscript, essentially for the continuous mode of cultivation (table and figure for each antioxidant compound). In fact, summarized tables and figures are suitable and give importance to the obtained results. The suggestion is to pool data of continuous culture:
Phenol, Alkaloids and Terpenoids concentrations at different light intensities in the same table.
Phenol, Alkaloids and Terpenoids productivity at 39.63 kJ d-1 in the same figure.
* Same remark for Tocopherols and carotenoids concentrations and productivity; Phycocyanin and Allophycocyanin concentrations and productivity.
The combination of data will give better reading of the obtained results for each species of the studied microalgae and will facilitate the comparison of their production of antioxidants under continuous mode of cultivation.
- Authors must give, in discussion section, a comparison of antioxidants production under Batch and continuous modes of cultivation to better conclude about the best cultivation mode
- Authors should justify the microalgae species choice.
- References of the protocols used for determination of different antioxidants must be cited.
- Replications for all experiments were given in results part. In order to add clarity and to avoid looking around for how much replication was used; authors should add replication number to each experimental section.
- Section of statistical analysis carried out be authors is missing.
- Reference citation should be uniformed according to Authors instructions.
- Several sentences are unclear or miss described. Authors should revise the manuscript.
e.g. line 197: “….although tocopherols concentration in P. cruentum was 1.5 times than in S. platensis”
Author Response
Point 1: - Line 40: Authors must explain abbreviations the first time they are used.
Response1: PUFAs full meaning was included polyunsaturated fatty acids.
Point 2: Line 86: in vivo must be in italic form.
Response 2: in vivo in italic form was changed in vivo.
Point 3: Throughout the manuscript: Authors should check missing italics in the scientific name of microalgae species
Response 3: All scientific names of microalgae checked for italic form.
Point 4: Line 113 to Line 117: Given informations are suitable to be included in Materials and Methods section; since it describe the experimental design of the conducted batch cultivation mode.
Response 4: All information regarding to materials and methods Line 113-117 moved to Materials and Methods section: See lines 386-387.
Point 5: Line 135 to Line 142: Same remark. Information about changing cultivation mode of microalgae from batch to continuous mode must be included on Materials and methods section.
Response 5: Information about changing cultivation mode of microalgae from batch to continuous mode was included on Materials and Methods section: Line 135 to 137. Moved to 388-391.
Point 6: Table 3 and 4: Authors must also indicate significant differences of means within the same column with different superscript letters
Response 6: Superscript letters in each column have included in tables 3 and 4 and for all the tables.
Point 7: Table 5: Letters indicating differences must be on superscript form.
Response 7: Table 5: Letters indicating differences were modified to superscript form, different superscript letters were included in all columns.
Point 8: Authors should give the significance of “nd” cited in tables.
Response 8: nd significance is given “no determined”
Point 9: The form and presentation of figures should be improved and revised
Response 9. Presentation of figures were improved and revised
Point 10. Authors inserted many tables and figures in the manuscript, essentially for the continuous mode of cultivation (table and figure for each antioxidant compound). In fact, summarized tables and figures are suitable and give importance to the obtained results. The suggestion is to pool data of continuous culture:
Phenol, Alkaloids and Terpenoids concentrations at different light intensities in the same table.
Phenol, Alkaloids and Terpenoids productivity at 39.63 kJ d-1 in the same figure.
Response 10.1 Data of concentration at different light intensities of phenols, alkaloids, and terpenoids were summarized in one table 6
Response 10.2 Data of productivity at different light intensities of phenols, alkaloids, and terpenoids were summarized in figure 5 (A, B, C)
Point 11. Same remark for Tocopherols and carotenoids concentrations and productivity; Phycocyanin and Allophycocyanin concentrations and productivity.
The combination of data will give better reading of the obtained results for each species of the studied microalgae and will facilitate the comparison of their production of antioxidants under continuous mode of cultivation.
Response 11.1 Data of productivity of tocopherol at different light intensity were included in figure 6
Response 11.2 Data of productivity of carotenoids at different light intensity were included in figure 7
Response 11.3 Data of productivity of UA SOD at different light intensity were included in figure 8
Response 11.4 Data of productivity of PC and APC at different light intensity were included in figure 9
Point 12. Authors must give, in discussion section, a comparison of antioxidants production under Batch and continuous modes of cultivation to better conclude about the best cultivation mode
Response 12. Conclusion was improved to make sure the better mode of cultivation.
Point 13. - Authors should justify the microalgae species choice.
Response 13. Several students have been working their thesis with one or more species to produce not only antioxidants but also antibiotics, anticancer, lectins and others. We know these species produce such compounds, for sample : Santiago-Morales, I. S.; Trujillo-Valle, L.; Márquez-Rocha, F. J.; López Hernández, J. F., Tocopherols, Phycocyanin and Superoxide Dismutase from Microalgae: as Potential Food Antioxidants. Applied Food Biotechnoly 2018, 5, (1), 19-27. doi: 10.22037/afb.v%i.17884
Point 14- References of the protocols used for determination of different antioxidants must be cited.
Response 14. Reference of antioxidant determination reference was added and cited correctly.
Point 15.- Replications for all experiments were given in results part. In order to add clarity and to avoid looking around for how much replication was used; authors should add replication number to each experimental section.
Response 15. Replication were given in all tables and figures, in addition a statistical section in material and methods was added
Point 16- Section of statistical analysis carried out be authors is missing.
Response 16. Statistical section was added in material and methods
Point 17- Reference citation should be uniformed according to Authors instructions.
Response 17. Citation and reference were verified as the Authors instructions indicate
Point 18. Several sentences are unclear or miss described. Authors should revise the manuscript.
Response 18. Sentences along the manuscript were revised to clarify it.
Point 18.1. e.g. line 197: “….although tocopherols concentration in P. cruentum was 1.5 times than in S. platensis”
Response 18.1 sentences in line 197 and others were revised.

Reviewer 2 Report
Dear the Editor
The authors provided data for antioxidant production by continuous cultivation procedure using photobioreactor with microalgae. The results of optimization have also been demonstrated. Importantly, a unique distribution of antioxidant production in various species of microalgae was demonstrated. Cultivation conditions were well-described.
1) The authors are encouraged to describe why this collection of microalgae has been selected. In fact, it is interesting to see these data, but some audience, including this Referee, hardly understand why this study has been done. Do these authors aim to maximize the yield of some particular antioxidant by choosing continuous cultivation system? Or do these authors simply want to demonstrate that these are data obtained under experimental conditions described by Method section.
2) Although described in the text, comparison of two or more values in figures needs to be clearly indicated. Furthermore, the result needs to be clearly shown by an asterisk.
3) English needs to be edited by an external editing service. Unfortunately, there are many grammatical mistakes in this manuscript.
Author Response
Point 1: The authors are encouraged to describe why this collection of microalgae has been selected. In fact, it is interesting to see these data, but some audience, including this Referee, hardly understand why this study has been done. Do these authors aim to maximize the yield of some particular antioxidant by choosing continuous cultivation system? Or do these authors simply want to demonstrate that these are data obtained under experimental conditions described by Method section.
Response 1: Several students have been working their thesis with one or more species to produce not only antioxidants but also antibiotics, anticancer, lectins and others. We know these species produce such compounds, for sample : Santiago-Morales, I. S.; Trujillo-Valle, L.; Márquez-Rocha, F. J.; López Hernández, J. F., Tocopherols, Phycocyanin and Superoxide Dismutase from Microalgae: as Potential Food Antioxidants. Applied Food Biotechnoly 2018, 5, (1), 19-27. doi: 10.22037/afb.v%i.17884. This study proved that microalgae studied produce a set of antioxidant, but at different amount in each microalgae. One disadvantage of microalgae cultivation is the low cell density, it is necessary to use some strategies in order to enhance cell density, some have been studied by culture medium changes, strain selection and bioreactor engineering and operations improvements. In this work, we use operation strategy, continuous cultivation has been used for bacterial cell and product productivity enhancement, in microalgal continuous cultivation is difficult to obtain. Another new point is that light intensity alter antioxidant concentration and/or productivity.
Point 2: Although described in the text, comparison of two or more values in figures needs to be clearly indicated. Furthermore, the result needs to be clearly shown by an asterisk.
Response 2: Comparison in all tables and figures are clearly indicated. Results were clearly showed in tables and figures
Point 3: English needs to be edited by an external editing service. Unfortunately, there are many grammatical mistakes in this manuscript.
Response 3: English edition has been provide by MDPI English edition service.
Round 2
Reviewer 1 Report
Authors have adequately made the main modifications suggested.
They answered to the most of the questions and remarks of the Reviewers.
The main concerns of the manuscript have been solved.
Therefore, we consider the manuscript suitable for publication.
Reviewer 2 Report
Dear the Editor,
These authors properly addressed severel scientific concerns raised by the Referee in the revised manuscript.